# Non-synaptic signaling from cerebellar climbing fibers modulates Golgi cell activity

Angela K Nietz, Jada H Vaden[†], Luke T Coddington[†], Linda Overstreet-Wadiche*, Jacques I Wadiche*

Department of Neurobiology, University of Alabama at Birmingham, Birmingham, United States

**Abstract** Golgi cells are the principal inhibitory neurons at the input stage of the cerebellum, providing feedforward and feedback inhibition through mossy fiber and parallel fiber synapses. In vivo studies have shown that Golgi cell activity is regulated by climbing fiber stimulation, yet there is little functional or anatomical evidence for synapses between climbing fibers and Golgi cells. Here, we show that glutamate released from climbing fibers activates ionotropic and metabotropic receptors on Golgi cells through spillover-mediated transmission. The interplay of excitatory and inhibitory conductances provides flexible control over Golgi cell spiking, allowing either excitation or a biphasic sequence of excitation and inhibition following single climbing fiber stimulation. Together with prior studies of spillover transmission to molecular layer interneurons, these results reveal that climbing fibers exert control over inhibition at both the input and output layers of the cerebellar cortex.

DOI: https://doi.org/10.7554/eLife.29215.001

*For correspondence: lwadiche@ uab.edu (LO-W); jwadiche@uab. edu (JIW)

[†]These authors contributed equally to this work

Competing interests: The authors declare that no competing interests exist.

## Introduction

Information transfer between neurons predominantly occurs at synapses, where neurotransmitter released by the presynaptic cell activates postsynaptic receptors that are clustered at anatomically-defined specialized structures. Anatomical reconstructions of synaptic connectivity thus delineate pathways of information flow throughout the brain. Since monoamine neurotransmitters and GABA can signal through volume transmission (*Fuxe and Agnati, 1991*; *Overstreet-Wadiche and McBain, 2015*), functional connectivity is not strictly synonymous with anatomical connectivity. Yet most excitatory neurotransmission is mediated by the fast neurotransmitter glutamate that is tightly limited to the synapse by diffusion and reuptake by glutamate transporters (*Wadiche and Jahr, 2005*; *Tzingounis and Wadiche, 2007*). Though this argues that anatomical connectivity faithfully represents excitatory circuits, at some synapses glutamate can spill out of the synaptic cleft at sufficient levels to activate nearby receptors (*Asztely et al., 1997*; *Isaacson, 1999*; *Chalifoux and Carter, 2011*; *Szmajda and Devries, 2011*). This process, termed spillover, has been studied extensively at cerebellar synapses, where it augments conventional point-to-point synaptic transmission between neurons and enables glutamate signaling to Bergmann glia (*Bergles et al., 1997*; *Linden, 1997*; *Carter and Regehr, 2000*; *Mitchell and Silver, 2000b*; *DiGregorio et al., 2002*; *Mitchell and Lee, 2011*; *Tsai et al., 2012*; *Coddington et al., 2014*). Despite the abundance of spillover transmission at cerebellar synapses, it is unclear how glutamate spillover contributes to Golgi cell (GoC) activity.

Cerebellar climbing fibers (CFs) establish powerful synapses with Purkinje cells (PCs) comprised of several hundred anatomically defined release sites (*Palay and Chan-Palay, 1974*). Each site releases multiple vesicles per action potential to generate a high glutamate concentration that overwhelms glutamate transporters, resulting in spillover to neighboring molecular layer interneurons (MLIs) in

the absence of an anatomical synaptic connection (*Palay and Chan-Palay, 1974*; *Wadiche and Jahr, 2001*; *Szapiro and Barbour, 2007*; *Mathews et al., 2012*; *Coddington et al., 2013*). Remarkably, the magnitude of spillover-induced AMPA and NMDA receptor-mediated depolarization is sufficient to trigger MLI spiking, and can modulate PC excitability through feed-forward inhibition (*Szapiro and Barbour, 2007*; *Mathews et al., 2012*; *Coddington et al., 2013*; *Coddington et al., 2014*). Thus, glutamate spillover signaling from CFs is sufficient to shape network dynamics even in the absence of anatomical connectivity.

GoCs are spontaneously active GABAergic interneurons providing the primary source of inhibition to granule cells. Early work suggested that CFs innervate GoCs (*Scheibel and Scheibel, 1954*; *Marr, 1969*), an idea supported by subsequent in vivo studies showing transient suspension of GoC tonic firing following CF stimulation (*Schulman and Bloom, 1981*; *Xu and Edgley, 2008*). However, a recent study failed to find structural evidence of synaptic contacts between CFs and GoCs (*Galliano et al., 2013*), and it is unclear how synaptic signaling by glutamatergic CFs would suppress GoC activity. Here, we show that glutamate spillover from CFs controls GoC activity with a distinct temporal profile in comparison to synaptic transmission from mossy fibers and parallel fibers, and can suppress GoC spiking by mGluR2 activation. These results reconcile previously incongruous findings obtained by anatomical mapping and functional studies, and combined with previous work show that CFs influence inhibition at all levels of the cerebellar cortex (*Jörntell and Ekerot, 2003*; *Szapiro and Barbour, 2007*; *Mathews et al., 2012*; *Coddington et al., 2013*).

## Results

### CFs signal to GoCs through spillover transmission

We recorded from over 200 GoCs in the granule cell layer of mouse cerebellar slices (*Dieudonne, 1998*; *Forti et al., 2006*; *D'Angelo, 2008*; see Material and methods and *Figure 1—figure supplement 1*). Parallel fiber (PF) terminals form synapses on apical GoC dendrites that extend into the molecular layer (ML) whereas mossy fiber (MF) terminals establish synapses on basal GoC dendrites in the granule cell layer (*Dieudonne, 1998*). We used focal stimulation in the molecular layer (while blocking GABAergic and glycinergic inhibition with picrotoxin, CGP55845 and strychnine) to trigger EPSCs with fast kinetics (*Figures 1Ai, D and E*; 137 ± 20 pA; n = 17) and facilitating paired-pulse ratios (*Figure 1Ai and F*), consistent with PF synapses (*Dieudonne, 1998*; *Beierlein et al., 2007*). Alternatively, focal stimulation in the white matter or granule cell layer yielded EPSCs with slightly faster kinetics (*Figures 1Bi, D and E*; 124 ± 16 pA; n = 14) and no short-term plasticity (*Figure 1Bi and F*), consistent with MF inputs (*Kanichay and Silver, 2008*; *Chabrol et al., 2015*). Both PF and MF EPSCs exhibited gradual increases in amplitude with increasing stimulus intensity (*Figure 1Ai and 1Bi*). In marked contrast, focal stimulation near Purkinje cell soma evoked all-or-none EPSCs in GoCs with kinetics slower than either PF- or MF-EPSCs (*Figures 1Ci, D and E*; 34.3 ± 4.2 pA; n = 11). Slow Purkinje cell layer-evoked EPSCs also exhibited robust paired-pulse depression (PPD), unlike PF- and MF-EPSCs (*Figure 1Ci and F*). NBQX (5 µM) application equally blocked the response from all three afferent pathways, indicating that EPSCs were mediated by AMPARs (PF: inhibition to 8.5 ± 4.2%, MF: 7.0 ± 3.2%, CF: 8.3 ± 0.4%; n = 3–6, p>0.99 for each comparison, ANOVA, data not shown). Based on the all-or-none response, strong PPD and the slow rise and decay kinetics of EPSCs, we hypothesized that stimulation near PC somata evokes EPSCs arising from nearby climbing fibers (CFs) that course through the granule cell layer en route to making synapses on PC dendrites and spines, similar to what is observed in molecular layer interneurons (MLIs, *Szapiro and Barbour, 2007*; *Mathews et al., 2012*; *Coddington et al., 2013*).

The actions of glutamate outside the synaptic cleft are highly sensitive to glutamate uptake blockade, such that the glutamate transport blocker TBOA strongly potentiates CF-mediated spillover to MLIs (*Bergles and Jahr, 1997*; *Szapiro and Barbour, 2007*; *Mathews et al., 2012*; *Coddington et al., 2013*). To assess whether the slow CF-GoC EPSC is due to glutamate spillover from distant release sites, we tested the sensitivity to glutamate uptake inhibition. Whereas the amplitudes of PF- and MF-evoked EPSCs were unaffected by TBOA (*Figure 1Aii, Bii, and G*), the slow Purkinje cell layer-evoked EPSC was significantly potentiated by TBOA (*Figure 1Cii and G*). While TBOA had little effect on the decay of the PF- and MF-evoked EPSC, the Purkinje cell layer evoked EPSC was robustly prolonged (PF: 106 ± 6%; MF = 111 ± 15%; CF = 257 ± 30%; n = 15, 9,

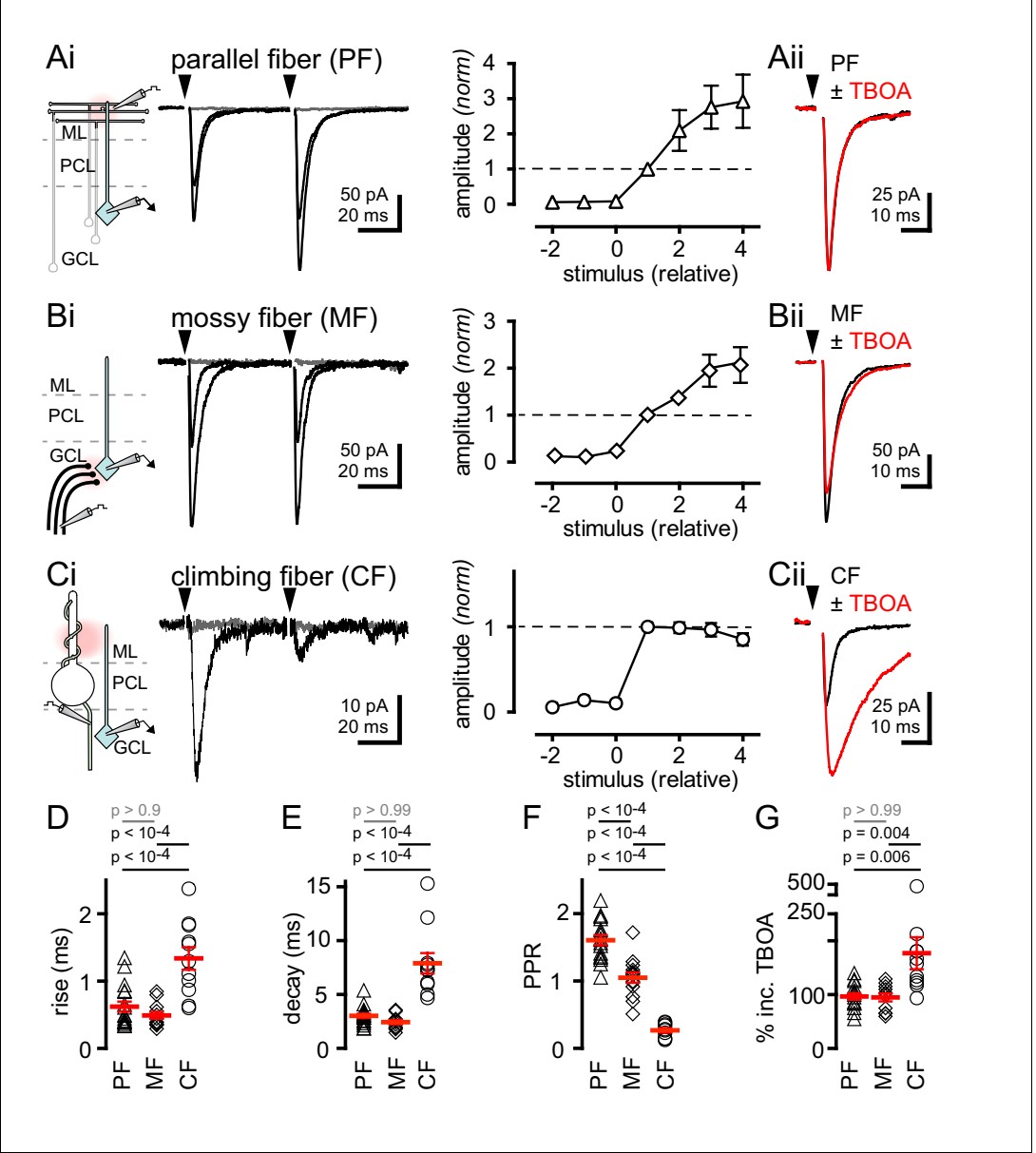

**Figure 1.** Synaptic- and spillover-mediated EPSCs onto cerebellar GoCs. . (**Ai**, left and middle) Parallel fibers (PFs) were stimulated with an electrode placed in the molecular layer (ML). Superimposed PF-GoC EPSCs in response to increasing stimulus intensity and paired-pulse (50 ms) stimulation. Gray trace is the average subthreshold response. Arrows denote stimulation. (**Ai**, right) Normalized PF-GoC EPSCs (n = 9, triangles) with increasing stimulus intensity relative to the first supra-threshold response (dotted line). (**Aii**) Inhibition of glutamate uptake (TBOA; red) does not affect the amplitude or time-course of PF-GoC EPSCs. Holding potential = −60 mV. (**Bi**, left and middle) Mossy fibers (MFs) were stimulated with an electrode placed in the white matter, below the granule cell layer (GCL). Superimposed MF-GoC EPSCs in response to increasing stimulus intensity and paired-pulse (50 ms) stimulation. (**Bi**, right) Normalized MF-GoC EPSCs (n = 7, diamonds) with increasing stimulus intensity relative to the first supra-threshold response (dotted line). (**Bii**) Inhibition of glutamate uptake (TBOA; red) does not affect the amplitude or time-course of MF-GoC EPSCs. (**Ci**, left and middle) Climbing fiber (CF) was stimulated with an electrode placed below the Purkinje cell layer (PCL). Superimposed CF-GoC EPSCs in response to increasing stimulus intensity and paired-pulse (50 ms) stimulation. (**Ci**, right) Normalized CF-GoC EPSCs (n = 13, circles) with increasing stimulus intensity relative to the first supra-threshold response. (**Cii**) Inhibition of glutamate uptake (TBOA; red) increases the peak amplitude and slows the kinetics of CF-GoC EPSCs. (**D**) rise-times, (**E**) decay-times, and (**F**) paired-pulse ratios following PF- (triangles), MF- (diamonds), and CF- (circles) stimulation. Red horizontal bars represent the mean ± SEM. PF: rise = 0.62 ± 0.08 ms, decay = 3.0 ± 0.2 ms, PPR = 1.6 ± 0.07; n = 17 each. MF:

*Figure 1 continued on next page*

*Figure 1 continued*

rise = 0.49 ± 0.05 ms, decay = 2.4 ± 0.2 ms, PPR = 1.05 ± 0.08; n = 14, 12, and 11. CF: rise time = 1.3 ± 0.2 ms, decay = 7.9 ± 0.9 ms, PPR = 0.27 ± 0.03; n = 11, 11, and 10). (G) Summary of % TBOA (50 μM) peak amplitude increase following PF- (triangles), MF- (diamonds), and CF- (circles) stimulation. PF: 97 ± 6%; MF: 95 ± 7%, CF: 177 ± 30%; n = 15, 11, and 11.

DOI: https://doi.org/10.7554/eLife.29215.002

The following figure supplements are available for figure 1:

**Figure supplement 1.** Electrophysiological and visual identification of cerebellar Golgi cells (GoC).
DOI: https://doi.org/10.7554/eLife.29215.003

**Figure supplement 2.** NMDA-receptor-EPSCs.
DOI: https://doi.org/10.7554/eLife.29215.004

and 11, PF v MF: p>0.99, PF v CF: $p<10^{-4}$, MF v CF: $p<10^{-4}$, ANOVA, data not shown). Finally, we tested whether NMDA receptors sensed glutamate spillover by voltage clamping GoCs at +40 mV by pharmacologically isolating NMDARs. Indeed, single CF-stimulation was sufficient to evoke CPP-sensitive currents suggesting that under depolarizing conditions NMDARs may contribute to CF-evoked responses (*Figure 1—figure supplement 2*). Together these results suggest that the slow Purkinje cell layer-evoked EPSC is a result of CF signaling to GoCs through glutamate spillover transmission. The slow kinetics, strong PPD and high sensitivity to TBOA establish criteria for distinguishing CF-mediated *spillover* from *synaptic* PF- and MF-evoked EPSCs.

## ChR2 stimulation of CF-mediated spillover and multiple CF recruitment

Next, we sought to determine whether GoCs could sense spillover from multiple CFs. We optogenetically activated CFs using mice that express ChR2 driven by the endogenous promoter/enhancer elements of the corticotropin releasing hormone (CRH) locus that targets a subset of inferior olivary neurons (*Sawada et al., 2008*; *Taniguchi et al., 2011*). CF afferents were identified by tree-like axonal arbors in the molecular layer expressing EYFP tagged ChR2 (*Figure 2A*). Because ChR2 expression was also evident in subsets of MF terminals, we isolated CF activation by targeting light to the molecular layer. Brief pulses of light (1–2 ms; 455 nm) generated all-or-none EPSCs onto GoCs with amplitudes and strong PPD similar to electrical stimulation near PCs (*Figure 2A* inset, *Figure 2B*, also see *Figure 2—figure supplement 1*). Furthermore, the kinetics of light-evoked EPSCs were similar to EPSCs evoked by electrical stimulation near PCs (rise: 1.7 ± 0.3 ms, decay: 8.0 ± 0.9 ms; n = 15 and 11, rise p=0.22 and decay p=0.96, unpaired t-tests, not shown), illustrating that ChR2 can be used to evoke CF spillover to GoCs in CRH-ChR2 mice.

The glutamate concentration at the CF-PC synapse and resulting spillover can be modulated through changes in release probability (*Dittman and Regehr, 1998*; *Wadiche and Jahr, 2001*; *Rudolph et al., 2011*). Thus, we tested if we could increase glutamate spillover onto GoCs by altering release probability. Modestly increasing extracellular calcium from 2.0 to 2.5 mM increased the average CF-GoC EPSC amplitude while prolonging the EPSC decay without altering the rise-time. Consistent with the increase in EPSC amplitude resulting from higher release probability, the PPR of ChR2-evoked EPSCs was reduced in 2.5 $Ca^{2+}$ (*Figure 2—figure supplement 2A*). However, increasing calcium concentration did not alter the sensitivity to TBOA (*Figure 2—figure supplement 2B*), suggesting that the increase in spillover signaling was not sufficient to alter the regulation of the extracellular [glutamate] profile by transporters.

Interestingly, in a subset of GoCs (11 of 23 cells), we found a step-wise increase in current amplitude with increasing light intensity (*Figure 2C*), suggesting that high intensity light stimulation was sufficient to recruit multiple active CFs (*Mathews et al., 2012*). In these cells, EPSCs maintained CF characteristics including strong PPD (0.15 ± 0.03 and 0.09 ± 0.02 for single and multiple CFs, respectively; n = 11, p=0.09 paired t-test) and slow kinetics (rise time: 1.7 ± 0.3 ms and 1.6 ± 0.23 ms; for single and multiple CFs, respectively; n = 11, p=0.79, paired t-test, not shown). Together these data show that electrical and optogenetic stimulation evoke similar spillover EPSCs and that GoCs can sense spillover from multiple CFs.

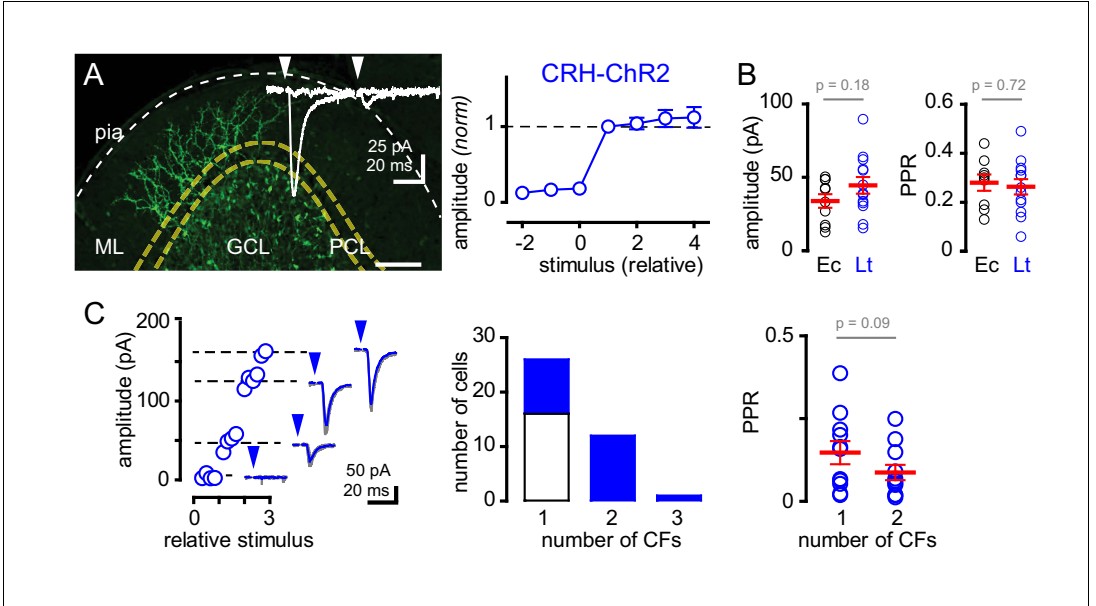

**Figure 2.** Recruitment of multiple climbing fibers with CRH-ChR2 stimulation. (**A**) Confocal Z-projection showing CFs expressing EYFP-tagged ChR2 in the ML from a parasagittal section of lobule III. Yellow and white dotted lines indicate boundaries of the PCL and pial surface, respectively. Inset shows representative CRH ChR2-EPSCs with strong depression following paired (50 ms inter stimulus) light stimulation. (**A**, right) Light-evoked (blue circles) EPSCs (n = 8) showed all-or-none behavior with increasing light intensity similar to PCL electrical stimulation (see *Figure 1Ci* for comparison). (**B**) The peak amplitude and PPR are similar with either electrical- (Ec; black circles) or light- (Lt; blue circles) stimulation. Light-evoked amplitude: 44 ± 6 pA and PPR: 0.26 ± 0.03; n = 13. (**C**, left) Example plot showing the recruitment of three CFs with increasing light intensity. Each discrete current measure (dotted line with EPSC) represents a putative CF. (**C**, middle) Summary graph showing frequency distribution of GoC receiving a discrete number of CFs. Light-evoked responses are shown in blue. On average light-stimulation can recruit 1.7 CFs. (**C**, right) Activation of multiple CFs onto GoCs does not change the PPR (1CF: 0.15 ± 0.03, 2CF: 0.09 ± 0.02, n = 11).

DOI: https://doi.org/10.7554/eLife.29215.005

The following figure supplements are available for figure 2:

**Figure supplement 1.** CF-PC light stimulation.

DOI: https://doi.org/10.7554/eLife.29215.006

**Figure supplement 2.** CF-GoC spillover EPSCs are sensitive to release probability.

DOI: https://doi.org/10.7554/eLife.29215.007

## Synaptic and spillover transmission generate distinct epochs of excitability

CF-MLI spillover signaling generates excitation and inhibition through spiking and di-synaptic feed-forward inhibition, respectively, demonstrating that non-synaptic communication can engage micro-circuits (*Mathews et al., 2012*; *Coddington et al., 2013*; *Coddington et al., 2014*). Accordingly, we tested how CF spillover signaling affects GoC spiking. Using voltage clamp recordings, we first confirmed the identity of PF, MF or CF-evoked EPSCs (as in *Figure 1*), and then assessed the consequences of stimulation of each pathway on GoC tonic firing. We constructed peristimulus probability histograms (PSHs) of GoC spiking (see Materials and methods). The intrinsic GoC spiking rate was comparable before PF, MF or CF stimulation (5.6 ± 0.8 Hz, 5.0 ± 0.6 Hz, 7.1 ± 0.4 Hz; n = 7, 6, and 25, PF v MF: p>0.99, PF v CF: p=0.25, MF v CF: p=0.09, ANOVA). Synaptic stimulation led to a transient and robust increase in the action potential (AP) frequency seen in raw traces and the PSHs (*Figure 3A and B*). PF stimulation increased the peak AP probability to 0.36 ± 0.07 within a narrow spike window (*Figure 3D and E*). Similarly, MF stimulation increased AP probability to 0.53 ± 0.09. In contrast, CF stimulation prompted a modest increase in the peak spike probability (*Figure 3C and D*) within a broader window that is consistent with the slow kinetics of spillover EPSCs

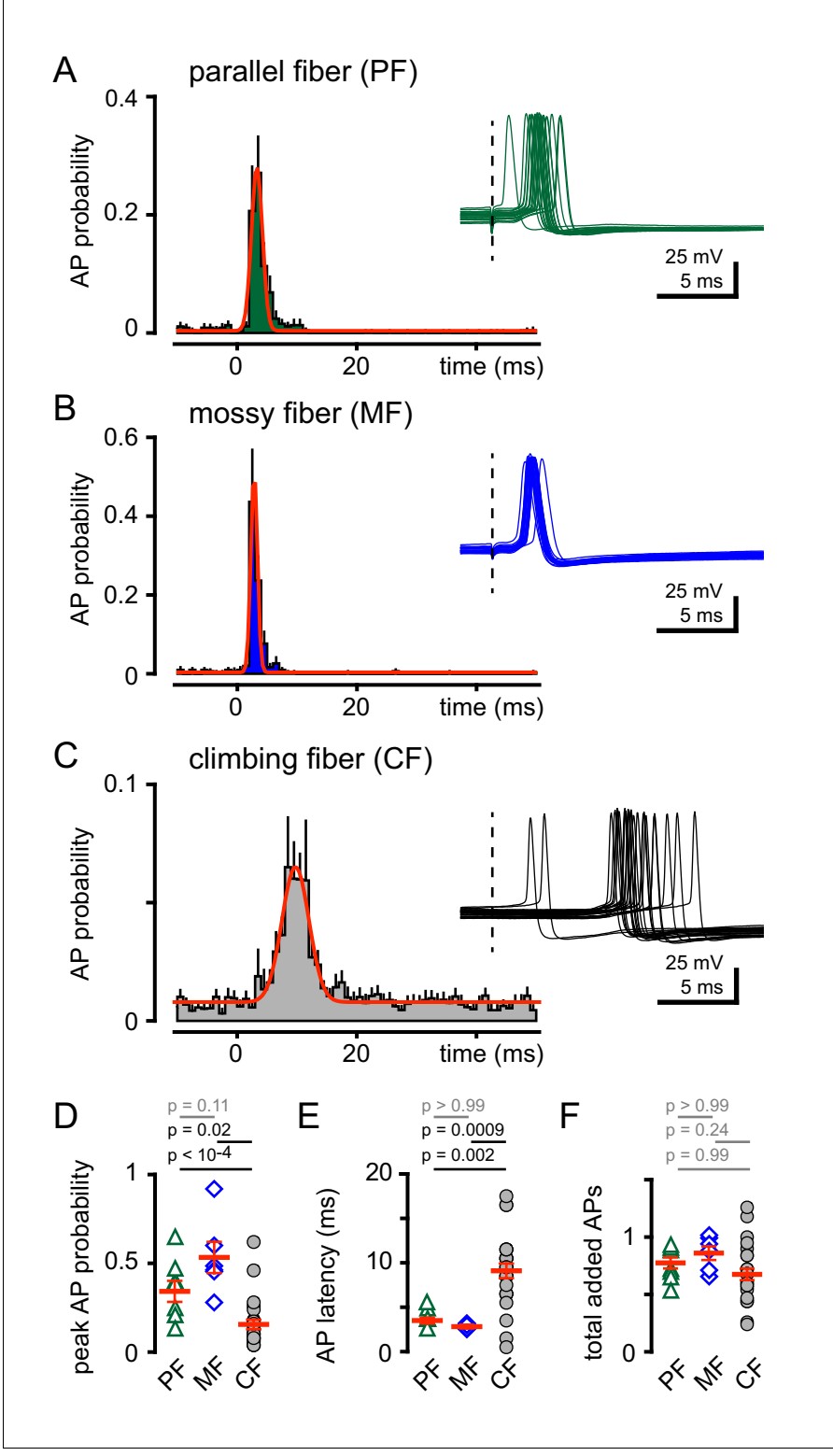

**Figure 3.** CF stimulation increases GoC spiking. (**A**) Average PF-GoC PSH (green) shows a rapid and large increase in the peak AP probability (0.36 ± 0.07; n = 7). The fit to a Gaussian distribution is overlaid in red (half-width: 0.53 ± 0.06 ms; n = 7). Inset illustrates the relatively low jitter in spike latency (3.6 ± 0.4 ms; n = 7) measured following PF stimulation (dotted line). (**B**) Average MF-GoC PSH (blue) shows a rapid and large increase in the peak AP probability (0.53 ± 0.09; n = 6). The fit to a Gaussian distribution is overlaid in red (half-width: 0.48 ± 0.07 ms; n = 6). Inset illustrates the relatively low jitter in spike latency (2.8 ± 0.2 ms) measured following MF

*Figure 3 continued on next page*

*Figure 3 continued*

stimulation. (C) Average CF-GoC PSH (grey) shows a slower and smaller increase in the peak AP probability (0.16 ± 0.03; n = 25). The fit to a Gaussian distribution is overlaid in red (half-width: 2.1 ± 0.7 ms; n = 10). Inset illustrates the high jitter in spike latency (9.1 ± 0.8 ms; n = 25) measured following CF stimulation. (D) AP probability and (E) pike latency following PF- (green triangles), MF- (blue diamonds), and CF- (grey circles) stimulation. Red horizontal bars represent the mean ±SEM. (F) Average number of added spikes following PF- (green triangles, 0.75 ± 0.06), MF- (blue diamonds, 0.86 ± 0.06), and CF- (grey circles, 0.68 ± 0.05) stimulation. Red horizontal bars represent the mean ± SEM.

DOI: https://doi.org/10.7554/eLife.29215.008

(*Figure 3E*). Despite the distinct peak frequencies and temporal windows, a similar number of APs were added (measured over 40 ms) for all stimulated afferents (*Figure 3F*). Thus, all afferent activity is uniformly capable of recruiting GoC spiking but with distinct temporal profiles.

## Uptake inhibition reveals biphasic GoC spiking

To further investigate the consequences of CF signaling, we assessed GoC spiking over a longer time period. CF stimulation transiently increased AP probability in most GoCs, with high variability in the return to baseline spiking (*Figure 4A*). To confirm that spiking was due to glutamate spillover from distant CF release sites, we tested the sensitivity to glutamate uptake inhibition that robustly increased CF-EPSCs (see *Figure 1*). TBOA increased the peak spiking probability over 250% as well as increased the time window of spiking, consistent with spillover signaling (*Figure 4B*). Unexpectedly, TBOA subsequently reduced GoC AP probability to approximately 70% of baseline (*Figure 4B*, ~200 ms after CF stimulation, arrow). This suggests that CF stimulation can regulate GoC spiking in a biphasic manner with both the increase and decrease in excitability regulated by glutamate transporters.

## CF stimulation can suppress GoC firing

To further explore the potential for bi-phasic regulation of GoC spiking with glutamate transporters intact, we more closely analyzed the raster plots and PSHs of individual GoGs with only inhibition blocked (from *Figure 4A*). Individual GoCs showed high variability in patterns of CF-induced spiking, from rapid recovery to baseline within 10 ms (*Figure 5A*) to complete suppression of APs for tens of milliseconds (*Figure 5B*). To distinguish between the spike and spike-pause pattern of GoC spiking,

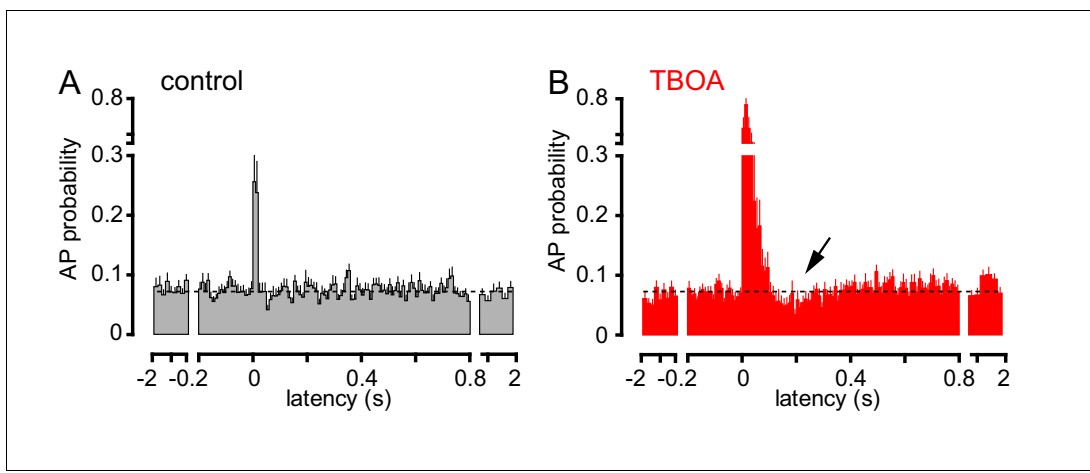

**Figure 4.** TBOA application reveals a biphasic effect of CF stimulation on GoC firing. (A) Average GoC spiking probability (20 ms bin) PSH following CF stimulation (t = 0 s) in control conditions (grey) or (B) in the presence of 50 μM TBOA (red). Dotted line indicates baseline AP probability. TBOA increased the peak spiking probability to 271 ± 22% and reduced spiking to 70.7 ± 7.2%,~200 ms after CF stimulation; n = 18, p<0.0001 and p=0.0007, respectively, paired t-test).

DOI: https://doi.org/10.7554/eLife.29215.009

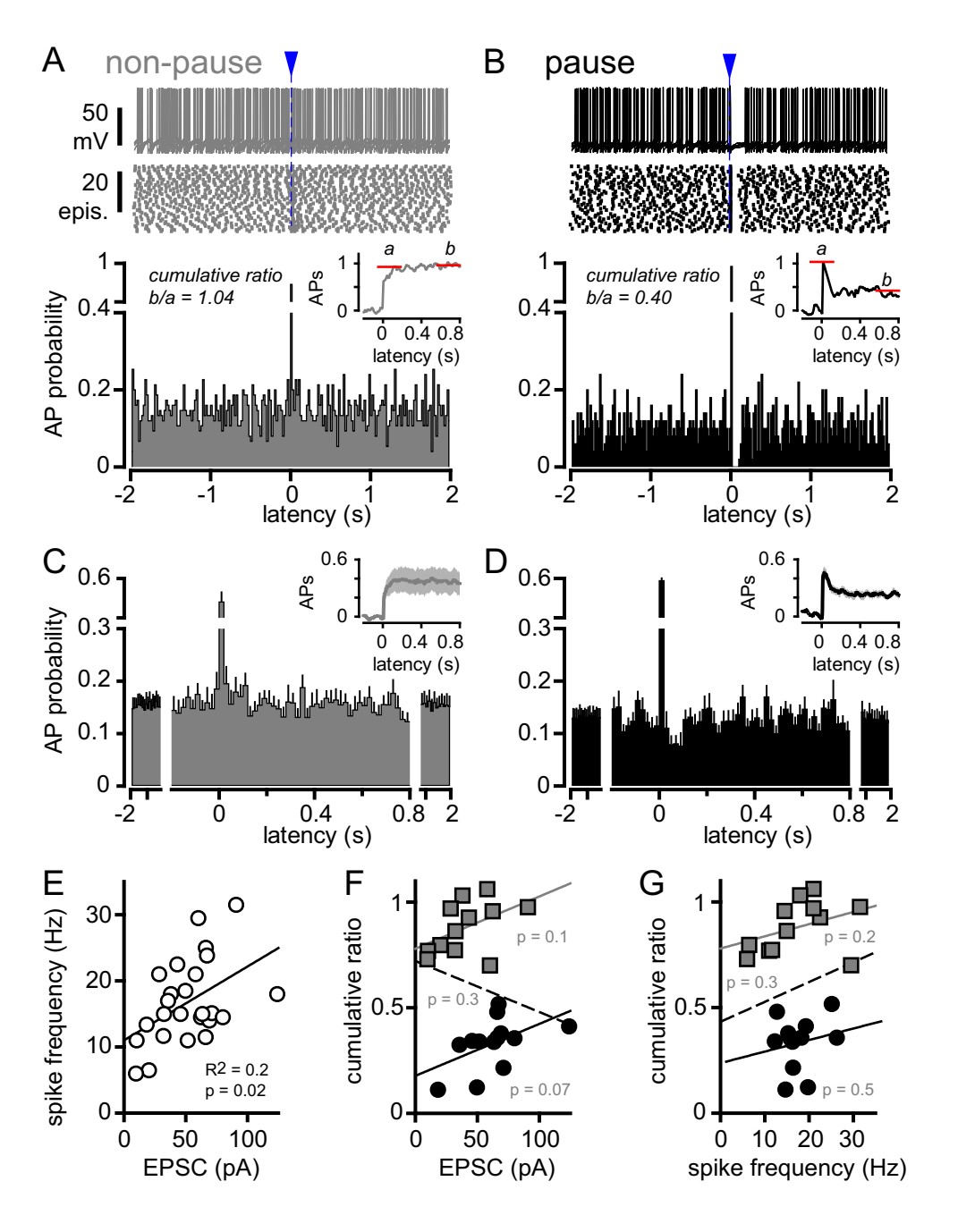

**Figure 5.** CF stimulation can suppress GoC firing. (**A**, top) Example traces (10 traces overlaid) and (middle) raster plot from a GoC that increases spiking in response to CF stimulation (blue arrow and dotted line) and immediately returns to baseline firing rate. (**A**, bottom) PSH of GoC spike probability (20 ms bin) with CF stimulation at time = 0 s. Inset shows the PSH integral to yield the cumulative spike probability (see text). Red lines illustrate the ratio of the steady state (**b**) to peak number of spikes (**a**) added with each stimulus to define a cumulative ratio (b/a). (**B**, top) Example traces (10 traces overlaid) and (middle) raster plot from a GoC that displays a suppression of GoC spiking following CF stimulation (arrow). (**B**, bottom) PSH of GoC spike probability (20 ms bin) with CF stimulation at time = 0 s. Inset shows the PSH integral to yield the cumulative spike probability with red lines illustrating the low cumulative ratio indicative of spike suppression. (**C and D**) Average PSHs for non-pausing (grey; cumulative ratio >0.65; n = 12 of 25) and pausing (black; cumulative ratio <0.65; n = 13 of 25) GoCs. Insets are the average cumulative spike probability for the non-pausing and pausing GoCs. (**E**) Summary plot showing that the average

*Figure 5 continued on next page*

*Figure 5 continued*
spike frequency is correlated with the EPSC amplitude (linear regression, n = 25 cells). (**F**) No correlation between the cumulative ratio and EPSC amplitude using all (dotted line), only non-pausing (grey squares and line), or pausing cells (filled circles and solid line). (**G**) No correlation between the cumulative ratio and average spike frequency using all (dotted line), only non-pause (grey squares and line), or pause cells tested (filled circles and solid line).
DOI: https://doi.org/10.7554/eLife.29215.010

we first integrated the PSH to yield a cumulative spike probability plot (*Figure 5A and B* insets) to quantify the number of action potentials added by CF stimulation (*Mittmann et al., 2005*; *Coddington et al., 2013*). The ratio of the steady state to the initial number of spikes added provided a relative measure of the inhibition that sometimes followed excitation (cumulative ratio, red lines in insets of *Figure 5A and B*; see Materials and methods) with a low cumulative ratio indicating spike suppression. There was considerable heterogeneity in the cumulative ratio across the population of GoCs, with a continuum between 0 and 1 that was not well fit by a model with two separate populations (not shown). We used this measure to quantify the biphasic nature of GoC spiking and classify cells as non-pause (spiking only) and pause (spiking followed by spike suppression) based on their response to GoC stimulation. We defined GoCs with a cumulative ratio >0.65 as non-pausing (*Figure 5A*) whereas GoCs with a ratio <0.65 were classified as pausing cells (*Figure 5B*). We averaged the PSHs and cumulative spiking plots of all non-pausing and pausing GoCs to highlight the distinct response patterns (*Figure 5C and D*). In GoCs classified as pausing cells, CF stimulation decreased the spike frequency from 6.7 ± 0.6 Hz to 3.3 ± 0.5 Hz (p<$10^{-4}$, paired t-test) for 93 ± 16 ms (n = 9 of 18 cells), which was equivalent to a reduction of the AP probability from 0.13 ± 0.01 to 0.07 ± 0.01 (*Figure 5D*).

Although there was no difference in the average peak AP probability between non-pause and pause GoC responses (0.42 ± 0.07 and 0.58 ± 0.07; n = 12 and 13, respectively, p=0.12, unpaired t-test), we wondered whether heterogeneity in CF-evoked pausing was related to variability in the CF-evoked EPSC. Not surprisingly, the CF-evoked average spike frequency (over 40 ms post CF-stimulation) was correlated with the CF-EPSC amplitude across all GoCs (*Figure 5E*, $R^2$ = 0.2; n = 25, p=0.02). However, we did not find a correlation between the cumulative ratio and the EPSC amplitude (*Figure 5F*) nor the cumulative ratio and the CF-evoked average spike frequency (*Figure 5G*). This suggests that GoC pauses did not result from intrinsic mechanisms that dictate the frequency of firing after depolarization, as occurs with PF- and MF-synaptic stimulation (see below).

## CF stimulation pauses GoC firing via postsynaptic mGluR2 activation

What generates the suppression of GoC excitability following CF stimulation? CF-spillover to MLIs generates a similar biphasic change in excitability resulting from sequential glutamate-mediated depolarization followed by GABAergic feedforward inhibition (*Mathews et al., 2012*; *Coddington et al., 2013*). However, inclusion of GABA receptor antagonists in our experiments precludes a contribution of feedforward inhibition. Alternatively, GoCs express mGluR2, a $G_i/G_o$ coupled glutamate receptor that reduces excitability by activating inwardly rectifying potassium channels (GIRKs; *Ohishi et al., 1994*; *Neki et al., 1996*; *Luján et al., 1997*; *Nakanishi, 2005*). CF-spillover could thus suppress GoC spiking via mGluR2-mediated GIRK activation, as occurs following strong or high frequency PF stimulation (*Watanabe and Nakanishi, 2003*).

We therefore tested whether the mGluR2 antagonist LY341495 altered CF-dependent GoCs spiking. We initially used simultaneous recordings in neighboring PCs to monitor if CF activation revealed an mGluR2-dependent pause in GoC spiking, although stimulus-induced spike resetting (see below) made the window of mGluR2-inhibition difficult to quantify across cells with heterogeneous spike patterns (*Figure 6—figure supplement 1*). We therefore used TBOA in addition to inhibitory blockers to enhance glutamate spillover. In the subset of GoCs showing CF-induced pauses (*Figure 6A*), TBOA increased the initial CF-evoked spiking probability (*Figure 6B and D*) as well as enhanced the duration of the subsequent pause period (from 93.3 ± 16 ms to 183 ± 23 ms; n = 9, p=0.01, paired t-test). The cumulative ratio trended to increase following TBOA application (0.6 ± 0.1) but was highly variable, likely due to the irregular effects on evoked spiking that could occlude the subsequent pause (*Figure 6D*; also see *Figure 1G* for variable TBOA effect on EPSCs).

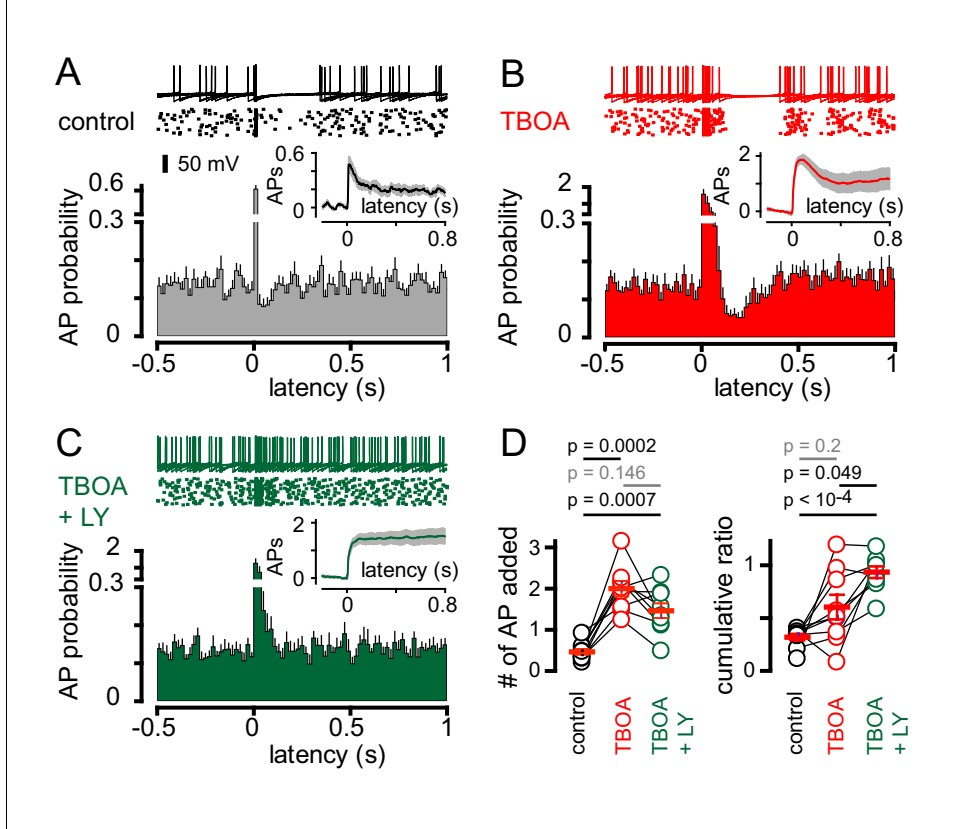

**Figure 6.** CF-mediated GoC pausing is dependent upon activation of postsynaptic mGluR2. (**A**, top) Example traces (10 traces overlaid) and raster plot (middle) from a GoC that displays a suppression of GoC spiking following CF stimulation (arrow) in control solutions. (**A**, bottom) PSH showing GoC spike probability constructed (20 ms bin) with CF stimulation at time = 0 s. Inset, shows the PSH integral to yield the cumulative spike probability (see text). (**B and C**) Same experimental paradigm as in (**A**) but in the presence of 50 µM TBOA (red), and in 50 µM TBOA +0.5 µM LY3414195 (green), respectively. (**D**, left) The number of added APs in response to CF stimulation increased from 0.46 ± 0.07 (black) to 2.0 ± 0.18 in the presence of TBOA (red) or TBOA +LY (1.47 ± 0.18 (green). (**D**, right) The average cumulative ratio in response to CF stimulation in control (black, 0.32 ± 0.03) was sustained in the presence of TBOA (red, 0.6 ± 0.1) and abolished by the application of TBOA +LY341495 (green, 0.93 ± 0.05).

DOI: https://doi.org/10.7554/eLife.29215.011

The following figure supplements are available for figure 6:

**Figure supplement 1.** CF-mediated GoC spike suppression is mGluR2 dependent.

DOI: https://doi.org/10.7554/eLife.29215.012

**Figure supplement 2.** Current injections reset GoC intrinsic activity.

DOI: https://doi.org/10.7554/eLife.29215.013

Inhibition of mGluR2s entirely blocked post-excitation pausing, returning the GoC spiking to baseline levels (*Figure 6C*; TBOA + LY341495 = 6.0 ± 0.9 and 6.7 ± 0.5 Hz, respectively; n = 9, p=0.28, paired t-test). Furthermore, mGluR2 inhibition did not significantly change the number APs added with CF-stimulation and resulted in a cumulative ratio comparable to that of non-pause GoCs (0.89 ± 0.04 n = 12 and 0.93 ± 0.06; n = 9, p=0.57, unpaired t-test and *Figure 6D*). Thus, CF-induced pausing results from spillover activation of mGluR2s. In the GoC subset showing a purely excitatory response (non-pause), application of TBOA enhanced post-stimulus spiking as expected and further application of LY341495 did not alter GoC activity (data not shown). Together, these results demonstrate that CF stimulation generates a rapid increase in GoC excitability which can be followed by a longer-lasting suppression of spiking. CF-induced inhibition happens in the absence of functional inhibitory connections, can be unmasked by uptake inhibition, and is dependent upon postsynaptic mGluR2 activation.

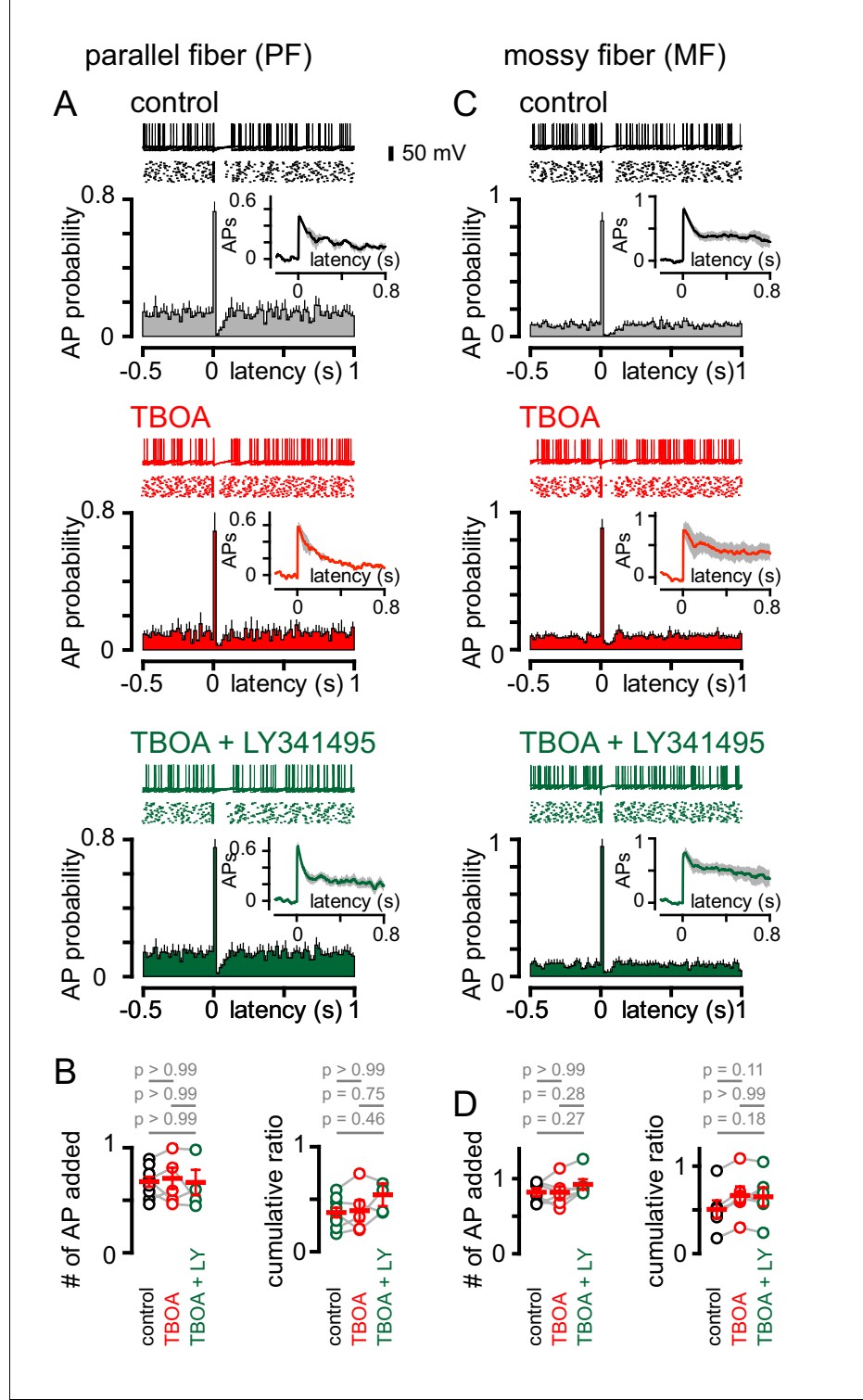

**Figure 7.** PF and MF stimulation produce an intrinsic pause in GoC firing. (**A**, top) Example traces (10 traces overlaid) and raster plot (middle) from a GoC that shows an increase followed by a short suppression in firing following PF stimulation. (**A**, bottom) Average PSH of GoC spiking probability (20 ms bins) with PF stimulation (time = 0 s). Inset, shows the integration of the PSH to yield cumulative spike probability plot. Data in control (lack), TBOA (50 μM, red) and TBOA +LY (50 μM + 0.5 μM, green). (**B**, left) Average number of added action potentials (APs) as a result of PF stimulation was measured at the peak of the cumulative probability plot in each condition. The average number of added APs was not different from control (black, 0.64 ± 0.05), in the presence of

*Figure 7 continued on next page*

*Figure 7 continued*
TBOA (red, 0.67 ± 0.1) or TBOA +LY (green, 0.62 ± 0.09; n = 5–7). (**B**, right) The average cumulative ratio after a single PF stimulus (black, 0.37 ± 0.05) was unaffected by the application of TBOA (red, 0.39 ± 0.1) and TBOA +LY (green, 0.54 ± 0.2; n = 5–7). (**C**, top) Example traces (10 traces overlaid) and raster plot (middle) from a GoC before and after MF stimulation. (**C**, bottom) Average PSH of GoC spike probability before and after MF stimulation (time = 0 s). Inset shows cumulative spike probability plot. (**D**, left) Average number of added APs as a result of MF stimulation was not different between control (black, 0.82 ± 0.05), TBOA (red, 0.82 ± 0.08) and TBOA +LY (green, 0.92 ± 0.07; n = 6). (**D**, right) The average cumulative ratio after a single MF stimulus (black, 0.54 ± 0.08) was unaffected by the application of TBOA (red, 0.66 ± 0.1) and TBOA +LY (0.65 ± 0.1; n = 6).
DOI: https://doi.org/10.7554/eLife.29215.014
The following figure supplement is available for figure 7:

**Figure supplement 1.** PF and MF cumulative ratio correlates with spike frequency.
DOI: https://doi.org/10.7554/eLife.29215.015

## PF and MF stimulation produce an intrinsic pause in GoC firing

We were surprised that a single CF stimulus was sufficient to trigger mGluR2-dependent GoC silencing. Previous work determined mGluR mediated inhibition of GoC excitability occurs only after strong high frequency PF-activation (*Watanabe and Nakanishi, 2003*). However, low frequency synaptic stimulation coupled with intrinsic mechanisms have been shown to efficiently reset GoC tonic spiking (*Vos et al., 1999*; *Forti et al., 2006*; *Solinas et al., 2007b*; *Kanichay and Silver, 2008*; *Vervaeke et al., 2010*). Indeed, we found a linear relationship between the spiking induced by direct current injection and the subsequent GoC spiking suppression suggestive of intrinsic spike resetting (*Solinas et al., 2007b*; *Figure 6—figure supplement 2*). However, blocking mGluR2 receptors had no effect, whereas blocking mGluR2 receptors selectively abolished the pause following CF stimulation (*Figure 6*). These results suggest that CF-evoked pausing of GoCs are not the result of phase resetting mechanisms that regulate pacemaking activity.

We next sought to determine the effect of single PF or MF stimuli on GoC firing. Stimulation of either afferent generated a biphasic pattern of GoC spiking, with the synaptically-evoked APs followed by a rapid and brief suppression of spiking (*Figure 7A and C*). In contrast to CF stimulation, the pattern of PF and MF induced spiking was unaffected either by TBOA or TBOA + LY341495 (*Figure 7A–7D*). The insensitivity of spiking to TBOA is consistent with the insensitivity of the PF and MF EPSCs to inhibition of glutamate transport (see *Figure 1*), and suggests that the pause in spiking reflects a resetting of tonic spike rate (or spike synchronization) due to intrinsic mechanisms that set the interspike interval. Supporting this idea, the PF and MF cumulative ratio was correlated with the spike frequency (*Figure 7—figure supplement 1*). These results highlight further differences in spillover and synaptic signaling to GoCs. Whereas both modes of transmission can trigger biphasic modulation of excitability, the underlying mechanisms mediating these changes are markedly different.

## Variability of CF-mediated GoC pausing is not due to heterogeneous mGluR expression

Heterogeneity in mGluR2/GIRK expression in GoCs has been postulated to determine an individual GoC's contribution to local network activity (*Rössert et al., 2015*). We tested whether the variability of GoC responses to CF spillover similarly result from a population of GoCs that do not express mGluR2 (*Neki et al., 1996*; *Luján et al., 1997*). We applied an exogenous mGluR agonist to GoCs to assess the presence of functional mGluR2, using TBOA to facilitate identification of pause/non-pause cells. In GoCs that lacked a CF-induced pause, assessed by a cumulative spike ratio >0.65 (*Figure 8A*), the mGluR2 agonist DCGIV generated a robust outward current that was blocked by LY341495 (*Figure 8A*). In GoCs with a cumulative ratio ≤0.65 (*Figure 8B*), DCGIV also generated an outward current consistent with GIRK channel activation that was subsequently blocked by LY341495 (*Figure 8B*). These data show that the heterogeneity of CF-evoked responses is not a result of receptor expression, since all the tested GoCs had a similar response to the exogenous mGluR2 agonist. Although GoCs that lack mGluR2 have been reported, they are located deep near the white matter (*Ohishi et al., 1994*), while we restricted our recordings to the middle and outer granule cell layer.

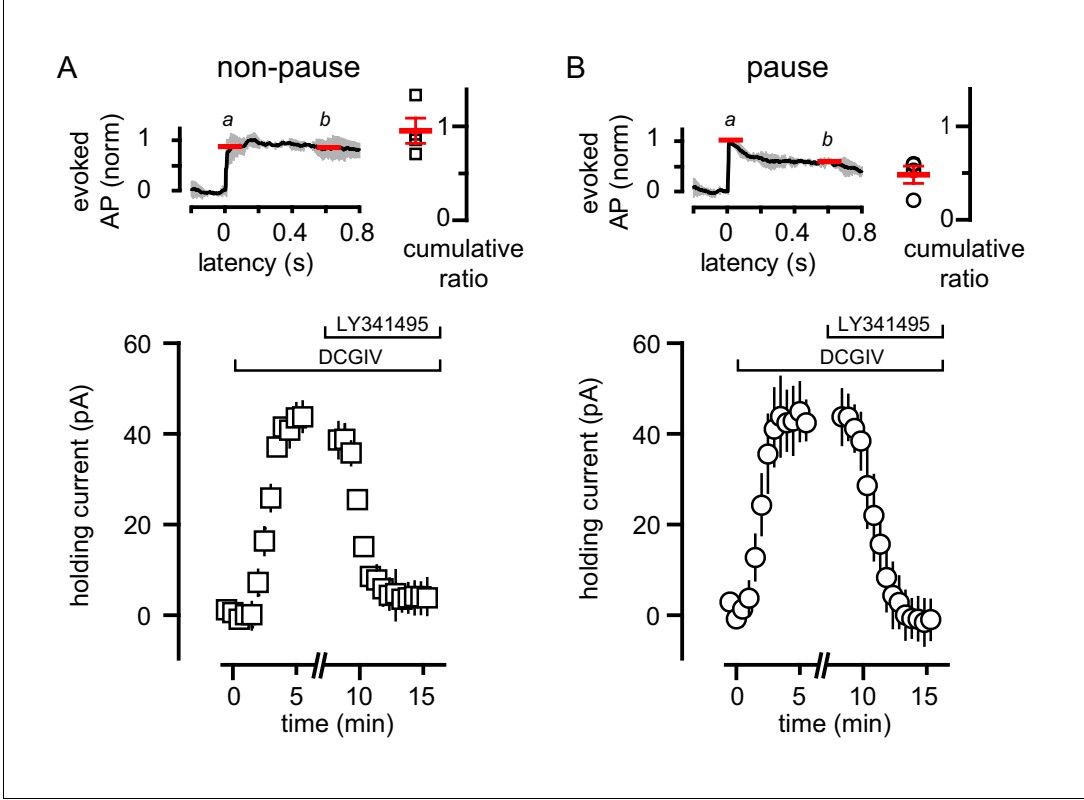

**Figure 8.** Variability of CF-mediated GoC pausing is not due to heterogeneous mGluR2 expression. (**A**, top) Normalized average number of added APs in TBOA (±SEM in grey) and cumulative ratio in response to CF stimulation in non-pausing GoCs (0.96 ± 0.13, n = 4). (**A**, bottom) The average holding current (40 ± 4 pA, n = 4; baseline normalized to 0 pA) before and after bath application of DCG-IV (1 µM) and LY341495 (0.5 µM) in non-pausing cells. (**B**, top) Normalized average number of added APs in TBOA (±SEM in grey) and cumulative ratio in response to CF stimulation in pausing (0.48 ± 0.09, n = 4) GoCs. (**B**, bottom) The average holding current (41 ± 9 pA, n = 4; baseline normalized to 0 pA) before and after bath application of DCG-IV (1 µM) and LY341495 (0.5 µM). The average holding current was not significantly different in the two groups (p=0.8, unpaired t-test).

DOI: https://doi.org/10.7554/eLife.29215.016

## GoC pausing is correlated with CF-evoked mGluR current

To test whether the degree of mGluR activation contributes to GoC response heterogeneity, we measured the CF-evoked mGluR2-mediated outward current associated with CF-evoked inhibition. In experiments where we recorded CF-evoked EPSCs and spiking in all three pharmacological conditions, we quantified the LY341495-sensitive current by assessing the charge difference between the TBOA and TBOA +LY341495 EPSCs (noting that paired-pulses at a 50 ms interval were used). As expected, GoCs with a large CF-evoked LY341495-sensitive current exhibited a robust pause following CF stimulation whereas GoCs with a small or undetectable CF-evoked LY341495-sensitive current did not exhibit pausing behavior (*Figure 9A*). Across nine GoCs there was a correlation between the LY341495-sensitive charge and the cumulative ratio, further confirming that CF-evoked pausing results from mGluR activation (*Figure 9B*). Altogether, these results suggest that the subcellular location of spillover signaling to GoCs, rather than differences in receptor expression give rise to the observed response heterogeneity.

## Discussion

Here, we show that cerebellar CFs signal to GoCs by glutamate spillover. Spillover generates direct excitation through activation of AMPARs, as well as inhibition through activation of mGluRs. The ability of glutamate spillover to inhibit GoC firing suggests that mGluR-mediated inhibition may

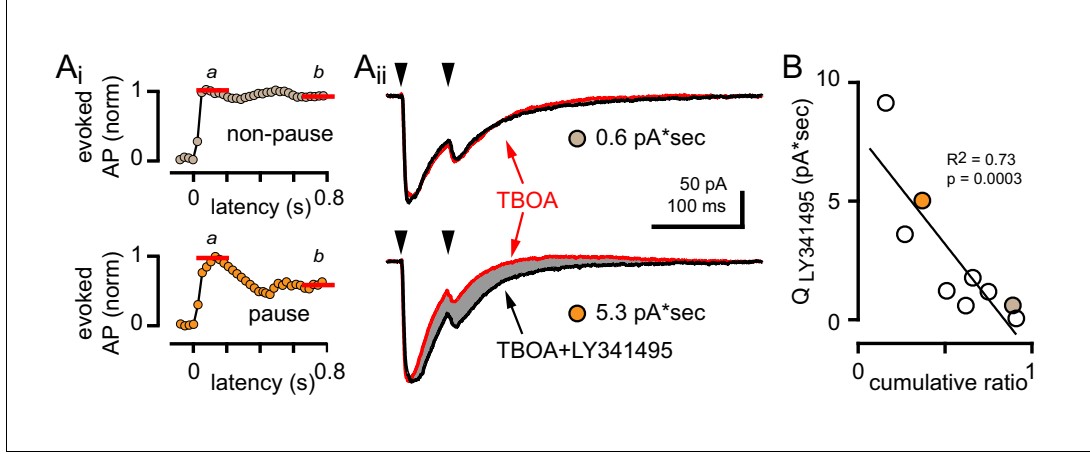

**Figure 9.** GoC pausing is correlated with a CF-evoked mGluR current. (**Ai**, top and bottom) Representative cumulative spike probability plot (normalized) for a non-pausing and pausing GoC in the presence of TBOA. (**Aii**, top and bottom) Example traces of CF-EPSCs (red, 50 μM TBOA) and TBOA +LY341495 (50 μM + 0.5 μM, black) for a non-pausing GoC and pausing GoC. Gray shaded region illustrates the amount of charge reduction observed with LY341495 application (0.6 pA*sec and 5.3 pA*sec, respectively for the non-pausing and pausing GoC shown). (**B**) Negative relationship between the charge blocked by LY341495 versus the cumulative ratio for each GoC. The quantity of CF-EPSC charge blocked by the application of LY341495 was measured by calculating the difference between the EPSC traces in TBOA +LY341495 and TBOA alone (n = 9). Colored circles indicate example cells from (Aii).

DOI: https://doi.org/10.7554/eLife.29215.017

contribute to the transient pauses in GoC tonic firing reported in vivo by stimulation of the inferior olive (**Schulman and Bloom, 1981**; **Xu and Edgley, 2008**). We show that antagonizing the mGluR2 receptor blocks the pause in GoC firing conferred by CF spillover, further supporting the importance of mGluR2 inhibition of GoC activity seen in vivo (**Holtzman et al., 2011**). Whereas CFs were suggested to innervate GoCs via thin collateral branches (**Shinoda et al., 2000**), recent work failed to find evidence for anatomical synaptic contacts between CFs and GoCs (**Galliano et al., 2013**). Our identification of spillover signaling may reconcile conflicting anatomical and functional results showing a lack of synaptic connections between CFs and GoCs yet functional consequences of CF stimulation.

## Cerebellar GoC afferent and efferent synaptic connectivity

GoCs receive information from all layers of the cerebellar cortex. GoCs extend short basal dendrites in the granule cell layer to sample MF activity (**Marr, 1969**; **DiGregorio et al., 2002**; **Ito, 2006**; **Kanichay and Silver, 2008**) and also extend 2–3 apical dendrites into the molecular layer to receive several thousand PF inputs (**Vos et al., 1999**; **Ito, 2006**; **Robberechts et al., 2010**). We expect that CF spillover signaling occurs on apical GoC dendrites in the ML, due to their proximity to CFs that innervate PC dendrites. Spillover signaling may enable integration of synaptic inputs in apical dendrites over a broader time window in contrast to the narrow window of integration of synaptic inputs arising in basal dendrites.

GoCs exert their influence in the GC layer at glomerular MF-GC synapses. Here, single mossy fiber boutons are contacted by ~50–100 GC dendrites and usually a single GoC axon (**Crowley et al., 2009**). The influence of GoCs is facilitated by the complex glomerular geometry that enables GABA released from GoCs to spillover to adjacent, non-synaptically coupled GCs (**Hamann et al., 2002**). The extensive ramification of the complex GoC axon also allows a single GoC to contact many glomeruli (**Ito, 2006**). Synaptic activation of GoCs generates feedforward phasic inhibition of GCs, and pacemaker activity contributes to tonic inhibition of GCs (**Brickley et al., 1996**). Thus CF-excitation of GoCs has the potential to influence the excitability of thousands of GCs.

In addition to their glutamatergic inputs, GoCs receive GABAergic inhibition from Lugaro cells (at a ratio of 10:1 LC:GoC; *Dieudonné and Dumoulin, 2000*; *Dumoulin et al., 2001*), cerebellar nuclear cells (*Ankri et al., 2015*), and other GoCs but not MLIs (*Dugué et al., 2009*; *Hull and Regehr, 2012*; *Szoboszlay et al., 2016*). Reciprocal connections between GCs and GoCs thus creates a traditional feedback inhibitory loop, that together with feedforward and tonic inhibition, maintain sparse population activity in the GC layer (*D'Angelo et al., 2013*). We speculate that reciprocal GABAergic synapses between GoCs or connections from Lugaro cells or recurrent PC-GoC connections (*Witter et al., 2016*) could contribute to CF-evoked pausing in vivo, a possibility that we did not test due to the difficulty of identifying CF-evoked EPSCs in the presence of intact inhibition.

## Stimulation of GoC afferents perturbs intrinsic firing by distinct mechanisms

We found that CF-mediated regulation of GoC spiking displayed distinct temporal properties compared to synaptic signaling from PF and MFs. Single PF and MF stimuli triggered rapid GoC spiking followed by a short-latency pause in tonic firing that was unaffected by mGluR2 antagonism. It is likely that this brief pause is due to the intrinsic pacemaker firing of GoCs that is synchronized across trials (*Vos et al., 1999*; *D'Angelo, 2008*). GoCs respond to afferent stimulation with a short burst of spikes that disrupts their intrinsic pacemaker activity (*Solinas et al., 2007b*, *2007a*). As a result, the pause seen after synaptic stimulation or current injection results from the synchronized reset in the pacemaker activity of the GoC.

In contrast, CF evoked suppression of GoC spiking depended on mGluR activation rather than intrinsic spike resetting. Both CF excitation and inhibition were enhanced by application of the glutamate transporter antagonist TBOA, but suppression of CF spiking was selectively blocked by mGluR2 antagonism, suggesting suppression results from mGluR-induced inhibition, similar to PF burst stimulation that activates postsynaptic mGluR2 on GoCs (*Watanabe and Nakanishi, 2003*; *Nakanishi, 2005*). In the intact brain, multiple active CFs may provide spillover to GoCs to produce a response under physiological conditions. In addition, spillover dependent mGluR-mediated inhibition may augment feedforward GABAergic synaptic inhibition imposed by other GoCs and Lugaro cells, although these possibilities remain to be tested.

mGluRs may be uniquely poised to sense glutamate spillover from active CFs. Unlike mGluR1/5 subunits that preferentially localize near excitatory synapses, mGluR2 is randomly distributed across the entire somatodendritic domain and axon of GoCs (*Luján et al., 1997*). The preferential activation of mGluR2 by single CF stimuli likely reflects the accessibility of spillover glutamate to reach sufficiently large regions of extrasynaptic GoC membranes to recruit detectable mGluR2-mediated GIRK activation. Because spillover is not confined to an anatomical post-synaptic site, its actions are largely regulated by glutamate clearance in the extracellular space and the proximity of postsynaptic receptors to the release sites. Our results showing that transport blockade enhances the duration of the mGluR2-mediated pause are consistent with this idea. In contrast, glutamate release at PF and MF synapses is likely constrained to peri-synaptic regions by a high density of glutamate transporters at excitatory synapses, such that extrasynaptic mGluR2 activation is controlled by the strength and duration of synaptic stimulation (*Watanabe and Nakanishi, 2003*). Our results confirm that glutamate released by single PF and MF stimuli was insufficient to activate mGluR2-mediated inhibition, even when glutamate transporters are blocked.

Golgi cells in vivo do not show excitation but exclusively show a pause in firing in response to climbing fiber stimulation (*Xu and Edgley, 2008*). It is possible that differences in experimental approaches between in vivo and in vitro studies contribute to this discrepancy in outcomes. We stimulated a single or a small number of CFs near recorded GoCs and identified CF-EPSCs by the well-known characteristics of CF spillover transmission (all-or-none EPSCs with PPD and TBOA sensitivity). Thus we 'selected' only GoCs that show AMPA receptor EPSCs - these recordings could be viewed as a 'paired' recording between a single CF and a single GoC. In vivo experiments use a protocol consisting of stimulation of the inferior olive, a site that is distant from recorded GoCs. We speculate that the discrepancy between in vivo and in vitro results reflects the difference in the probability of identifying a 'paired' recording between a particular CF and GoC in the two preparations. We expect that CFs can generate a range of GoC responses including excitation only, biphasic excitation-inhibition and inhibition only, depending on the complement of glutamate receptors responding to spillover. Since mGluRs have a higher affinity for glutamate than AMPARs (*Malherbe et al.,*

*2001*; *Traynelis et al., 2010*), it is possible that most GoCs in vivo exhibit solely an inhibitory response, but due to our selection criteria for confirming the CF source of glutamate, all of our GoC recordings have AMPA receptor components that will undoubtedly cause excitation. Just as extensive anatomical studies have detailed traditional synaptic connectivity, further work is needed to understand how release dynamics and tissue architecture might shape activation of extrasynaptic receptors to regulate functional connectivity.

### Implications for granule cell processing and heterogeneity

GoCs are responsible for regulating the responsiveness of GCs to MF afferent activity and thus setting the tone of information flow through the cerebellar cortex. The activity of GoCs sets the firing threshold as well as controls the timing of GC firing and subsequent information transfer to Purkinje cells (*Gabbiani et al., 1994*; *De Schutter et al., 2000*; *Vos et al., 2000*). Tonic and spillover inhibition from GoCs play a large role in determining the number of GCs able to respond to a given MF input (*Brickley et al., 1996*; *Rossi et al., 2003*; *Duguid et al., 2012*), which in turn helps generate a sparse representation that is thought to enhance the storage capacity of the cerebellum. Phasic inhibition generated by PF and MF activity serves to synchronize inhibition, and by extension GC activity generating center-surround inhibition (*Vos et al., 1999*; *D'Angelo and De Zeeuw, 2009*; *Vervaeke et al., 2010*).

Since GoCs regulate transmission at the mossy fiber-granule cell relay, CF input could serve to demarcate groups of GoCs depending on the type of response to CF input (with or without an mGluR-induced pause) to promote spatial filtering of mossy fiber signals (*Mitchell and Silver, 2000a*, *2000b*, *2003*). In addition, recent models have moved away from the granule cell layer being a simple spatial filter to emphasize its role in the timing of motor commands (*Yamazaki and Tanaka, 2009*; *Rössert et al., 2015*). In this context, GoCs set the time window for GC spiking and time is encoded through the sequential activation of populations of GCs. Indeed, results from behavioral assays in combination with lesioning studies have shown that disrupting the cerebellar cortical circuit leads to changes in learned response times while failing to abolish the learned response itself (*Perrett et al., 1993*; *Raymond et al., 1996*; *Prestori et al., 2008*). In the context of timing, spillover may serve to desynchronize cells within each layer of the cortex to promote efficient learning.

Given the regularity with which the cerebellar cortex is organized, it becomes necessary for cells to generate a multitude of responses to distinguish between stimuli (*Yamazaki and Tanaka, 2005*; *Rössert et al., 2015*). This requires significant heterogeneity and randomness in GCL activity patterns. Heterogeneity can arise from differential connectivity patterns between GCs and mossy fibers, as well as by recurrent inhibition from the GoC network which enhances sparse GC activity. An additional source of heterogeneity comes from receptor activation profiles. In recent cerebellar models, mGluR2 activity was suggested to increase the dynamic range of GC responses while preventing the circuit from entering an erratic state (*Rössert et al., 2015*). Thus, heterogeneity in CF-mediated mGluR activation on GoCs could provide as a source of variability in GCL activity.

Our results identify a new mechanism for regulating GoC activity at the cerebellar input stage. Here, conjunctive stimulation of CFs and peripheral inputs may lead to LTD at the MF-GC relay and/or plasticity at excitatory GoC synapses (*Xu and Edgley, 2008*; *Robberechts et al., 2010*). In the broader context of cerebellar function, it is possible that CF spillover allows local interneuron circuits to monitor input to PCs and regulate local circuit excitation and inhibition required to refine cerebellar-dependent actions. In the context of CFs providing an error signal, spillover signal to GoCs may relay privileged information at the input stage that is not encoded by mossy fibers. In this manner, CF spillover may enhance response complexity and regulate signaling through a form of transmission not constrained to synapses.

## Materials and methods

All experiments were conducted through protocols approved by the Institutional Animal Care and Use Committee of the University of Alabama at Birmingham

### Slice preparation

Mice (C57BL/6 or CRH-ires-CRE, Jackson Labs stock # 012704 x ChR2 (H134R)-EYFP, Jackson Labs stock # 012569) of either sex aged P16-28 were anesthetized via isoflurane inhalation or with 2, 2, 2-

tribromoethanol and intra-cardially perfused with ice cold cutting solution prior to decapitation. The cerebellum was quickly dissected into one of three ice cold cutting solutions (in mM): solution 1 (110 CholineCl, 7 MgCl$_2$, 2.5 KCl, 1.25 NaH$_2$PO$_4$, 0.5 CaCl$_2$, 25 glucose, 11.5 Na-Ascorbate, 3 Na-pyruvate, 25 NaHCO$_3$); solution 2 (130 KGluconate, 15 KCl, 0.05 EGTA, 20 HEPES, 25 Glucose with 2.5 µM R-CPP); solution 3 (85 NaCl, 75 Sucrose, 24 NaHCO$_3$, 25 Glucose, 4 MgCl$_2$, 2.5 KCl, 1.25 NaH$_2$PO$_4$, 0.5 CaCl$_2$ with 2.5 µM R-CPP). Solution 1 was typically used for voltage clamp experiments while solution 2 and 3 were used for current clamp experiments that assay tonic firing (*Hull and Regehr, 2012*; *Santhakumar et al., 2013*). Parasagittal slices from the cerebellar vermis (300 µM) were cut and incubated for 30 min in 35℃ ACSF in 119 or 125 NaCl, 26 NaHCO$_3$, 2.5 KCl, 1–1.25 NaH$_2$PO$_4$, 11 or 25 Glucose, 2–2.5 CaCl$_2$, 1–1.3 MgCl$_2$, and 2.5 µM R-CPP before being stored at room temperature. Recordings were performed in the same ACSF as used for incubation with R-CPP omitted for up to 5 hr post-slicing.

## Electrophysiology

Recordings were made from visually identified Golgi cells (GoCs). GoC identity was confirmed by the observation of action potential firing in either the cell-attached or whole-cell configurations and characteristic input resistance (see *Figure 1—figure supplement 1*; 185 ± 22 MΩ, n = 45). Recordings were made at ~32℃ maintained with an inline heater (Warner Instruments, Hamden, CT). Cells were visualized using a 60X water immersion objective on an Olympus BX51WI microscope equipped with infrared contrast optics (*Dodt et al., 2002*). Synaptic activity was recorded using a Multiclamp 700B amplifier and pClamp10 acquisition software (Molecular Devices, Sunnyvale, CA). Signals were filtered at 4–10 kHz and digitized at 20–50 kHz (Digidata 1440). Patch pipettes (BF150-086; Sutter Instruments, Novato, CA) were pulled on a Sutter P-97 (Sutter Instruments) horizontal puller to a resistance between 2.5 and 6 MΩ.

For voltage-clamp experiments, pipettes were filled with solutions containing (in mM): 140 CsMeSO$_3$, 15 HEPES, 0.5 EGTA, 2 TEA-Cl, 2 Mg-ATP, 0.3 Na-GTP, 10 phosphocreatine, 5 QX-314Cl, and 0.1 spermine. For current-clamp experiments, pipettes were filled with solutions containing (in mM): 150 KGluconate, 3 KCl, 10 HEPES, 0.5 EGTA, 3 Mg-ATP, 0.5 Na-GTP, 5 phosphocreatine (ditris), 5 phosphocreatine (disodium) or 90 KH$_2$PO$_4$, 10 KCl, 10 HEPES, 10 BAPTA, 4 MgCl$_2$, 0.4 Na-GTP, 2 Mg-ATP, 5 phosphocreatine (dipotassium) pH to 7.2 with KOH. Series resistance (R$_s$) was monitored by responses to a 5 mV voltage step and compensated to remain less than 10 MΩ. If R$_s$ changed significantly (≥20%) experiments were discarded. Current injection (20–100 pA) was sometimes used to maintain a baseline firing rate of 5–10 Hz. If action potential firing ceased or became erratic experiments were discarded.

Climbing fibers were stimulated using a theta glass pipette positioned near the Purkinje cell layer (100 µs, 20–100 µA). The stimulus intensity was adjusted to produce an all-or-none response with minimal parallel fiber (PF) or mossy fiber (MF) contamination. PFs or MFs were stimulated by positioning the theta glass pipette in the molecular layer or in the granule cell layer/white matter (>300 µm lateral to the recording pipette), respectively. PF- and MF- stimulus intensity was adjusted to keep EPSCs below 300 pA to minimize transmitter pooling (*Clark and Cull-Candy, 2002*). ChR2-expressing CFs were stimulated by positioning the epifluorescence-mounted LED (455 nm; Thorlabs, Newton, NJ) over the molecular layer. In most cases, the light was directed to the outer molecular layer knowing that it would spread to the middle and inner molecular layers to minimize activation of mossy fibers that sporadically express ChR2. For all current-clamp experiments we first confirmed that EPSCs evoked by putative CF stimulation displayed slow kinetics and profound paired-pulse depression; stimulation near the Purkinje cell layer or light stimulation that evoked EPSCs with fast kinetics and little synaptic depression (indicating contamination by PFs or MFs) were excluded from further analysis. Additional whole cell recordings were made from PCs in ChR2-expressing mice. Pipettes were filled with a solution containing in mM: 110 CsCl, 35 CsF, 10 Hepes, 10 EGTA, and 5 QX-314Cl. Recordings were made at −60 mV in the presence of 100 nM NBQX to reduce the size of CF-evoked EPSCs. In all cells tested, CF EPSCs were all-or-none in nature and no activation of multiple CFs onto individual PCs was observed.

## Immunohistochemistry and confocal imaging

In a subset of cells 0.2% biocytin was included in the intracellular solution to allow post-hoc morphological identification of GoCs. Slices were post fixed in 4% paraformaldehyde (PFA) for 24 hr. Free floating sections were washed with 1X PBS and rinsed with 0.3M glycine and 0.5% Triton-X 100. Slices were blocked in TBS containing 10% normal goat serum, 3% bovine serum albumin, 1% glycine, and 0.4% Triton-X 100 for 1 hr at room temperature. After the initial block, slices were incubated with Streptavidin conjugated Alexa 647 (1:1000; Invitrogen, Waltham, MA) overnight at 4°C. In cases of transgenic CRH-ires-CRE floxed ChR2-EYFP, slices were also incubated (overnight at 4°C) with rabbit anti-EGFP (1:1000; Invitrogen) to amplify staining of the EYFP-fused ChR2 protein. After staining, slices were rinsed with PBS and mounted with Vectashield anti-fade reagent (Vectorlabs, Burlingame, CA). Images of cells (Alexa 647) or CFs (ChR2-EYFP) were acquired using a 20X oil-immersion objective (0.85 NA) on an Olympus FluoView 300 confocal microscope using a 633 nm or 488 nm excitation wavelength. Images were processed using ImageJ software (NIH, Bethesda, MD).

## Data and statistical analysis

Recordings were analyzed using AxographX software (Axograph Scientific, Sydney, Australia). Changes to basal spontaneous action potential firing were quantified as in (*Coddington et al., 2013*). Briefly, peristimulus histograms (PSH) were computed and integrated. A linear fit to the baseline (200–300 ms) of the integral was extrapolated and subtracted from the integral to yield the cumulative spike probability plot. The ratio of the steady state (600–800 ms) to the initial number of spikes added (within 20 ms) provided a relative measure of the inhibition that sometimes followed the excitation with each stimulus (see *Figure 5*, insets). Average PSH and cumulative probability plots were calculated for cells with at least 20 individual episodes. Data are displayed as mean ±SEM and statistical significance was determined using paired or unpaired two-tailed Student's t-tests or ANOVAs followed by Bonferroni's multiple comparison test (GraphPad Prism, La Jolla, CA). In some figures, the values of the number of cells (n) tested changed if experiments were not carried out through all pharmacological treatments.

## Drugs

Picrotoxin (GABA$_A$R antagonist; 100 µM), NBQX (AMPAR antagonist; 5 µM), QX-314 (Na$^+$-channel blocker; 5 mM), and CGP55845 (GABA$_B$R antagonist; 2 µM) were obtained from Abcam (Cambridge, UK). R-CPP (NMDAR antagonist; 5–10 µM), LY341495 (mGluR2 antagonist; 0.5 µM), DCG-IV (mGluR2 agonist; 1 µM), and DL-TBOA (EAAT antagonist; 25–50 µM) were purchased from Tocris Bioscience (Minneapolis, MN). Strychnine (GlyR antagonist; 1 µM) and all other drugs and chemicals were obtained from Sigma Aldrich (St. Louis, MO) or Fisher Scientific (Pittsburgh, PA).

## Acknowledgements

We thank members of the Wadiche labs for helpful comments throughout this project and Mary Seelig for technical assistance. This work was supported by NIH F31NS093960 (AKN), NIH NS064025 (LOW) and NIH NS065920 (JIW).

## Additional information

### Funding

| Funder | Grant reference number | Author |
| --- | --- | --- |
| National Institute of Neurological Disorders and Stroke | F31 NS093960 | Angela K Nietz |
| National Institute of Neurological Disorders and Stroke | R01 NS064025 | Linda Overstreet-Wadiche |
| National Institute of Neurological Disorders and Stroke | R01 NS065920 | Jacques I Wadiche |

The funders had no role in study design, data collection and interpretation, or the decision to submit the work for publication.

## Author contributions

Angela K Nietz, Conceptualization, Formal analysis, Funding acquisition, Validation, Investigation, Visualization, Methodology, Writing—original draft, Writing—review and editing; Jada H Vaden, Conceptualization, Resources, Data curation, Formal analysis, Supervision, Funding acquisition, Visualization, Writing—original draft, Project administration, Writing—review and editing; Luke T Coddington, Conceptualization, Validation, Investigation, Writing—review and editing; Linda Overstreet-Wadiche, Conceptualization, Resources, Formal analysis, Supervision, Funding acquisition, Validation, Visualization, Methodology, Writing—original draft, Project administration, Writing—review and editing; Jacques I Wadiche, Conceptualization, Resources, Data curation, Formal analysis, Supervision, Funding acquisition, Validation, Visualization, Methodology, Writing—original draft, Writing—review and editing

## Author ORCIDs

Linda Overstreet-Wadiche, https://orcid.org/0000-0001-7367-5998
Jacques I Wadiche, http://orcid.org/0000-0001-8180-2061

## Ethics

All experiments were conducted through protocols approved by the Institutional Animal Care and Use Committee (IACUC protocol 08767) of the University of Alabama at Birmingham.

## Decision letter and Author response

Decision letter https://doi.org/10.7554/eLife.29215.019
Author response https://doi.org/10.7554/eLife.29215.020

# Additional files

## Supplementary files

• Transparent reporting form
DOI: https://doi.org/10.7554/eLife.29215.018

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
