## [Decision Letter]

Thank you for submitting your article "Non-synaptic signaling extends the influence of cerebellar climbing fibers" for consideration by *eLife*. Your article has been reviewed by three peer reviewers, and the evaluation has been overseen by a Reviewing Editor and Eve Marder as the Senior Editor. The reviewers have opted to remain anonymous.

The reviewers have discussed the reviews with one another and the Reviewing Editor has drafted this decision to help you prepare a revised submission. While all agreed that the data are of high quality and the results novel, the reviewers shared a common concern regarding the presentation. In particular, the caveats related to the specific experimental conditions used (NMDA/GABA receptor block, TBOA) were not adequately addressed, and along the same lines, the conclusions were overstated. We hope you will be able to submit the revised version within two months. If you decide to submit a revision, we will again seek the advice of the reviewers before a final decision is made.

Summary:

The authors show in their in vitro studies of the cerebellar microcircuit that release of glutamate during artificial climbing fiber activation can spill-over onto Golgi cells activating ionotropic and metabotropic glutamate receptors leading, respectively, to excitation then inhibition. The inhibition of Golgi cells occurs through activation of mGluR2, and this is consistent with previously published in vivo recordings showing a pause in Golgi cell firing following climbing fiber stimulation, suggesting an interesting underlying mechanism for inhibition at the input stage of the circuit.

We think you need to make the title more accessible to the broad readership of *eLife*, perhaps by including something about potential function of this part of the circuit, or likely consequence of your findings that would not have been previously expected. A more specific title that raises GoCs or the GC layer would seem appropriate.

Essential revisions:

1) In several instances the authors conclude that climbing fiber activity provides a biphasic (excitatory followed by inhibition) control over Golgi cell spiking. However, this statement is mainly true in the presence of the glutamate transport blocker, TBOA. In the absence of TBOA, roughly half of the cells show a purely excitatory response (Figure 5). The authors need to acknowledge that the statement of biphasic response was only true in a subpopulation of cells, and discuss the caveats of these methodological details in terms of their interpretation of results in the broader context of cerebellar function. How do the authors imagine that these mechanisms are engaged under physiological circumstances, i.e. without TBOA/Picro/CGP, etc?

2) Could the pause be driven by the higher peak spiking probability that occurs in TBOA? In Figure 4, the example shows a peak spiking probability of ~0.3 which increases to ~0.8. This ambiguity could be ruled out by calculating PSHs with different current injections that mimic the increases in peak probability and comparing the post-peak pauses.

3) In vivo recordings of Golgi cells following climbing fiber stimulation do not show excitation and always have a pause in firing (Xu and Edgley, 2008), whereas these data show a consistent excitation and a variable pause in firing. The authors should address this discrepancy between the in vivo and in vitro data.

4) The rationale for this study is a bit overstated. In particular, it is not clear how the results of the present manuscript provide more than additional examples of how high release synapses such as CFs can access distant receptor pools, including mGluRs. Further, how the findings increase our understanding of how extrasynaptic signaling contributes to local circuit processes could be clarified. For example, does spillover transmission to mGluRs combine with MF inputs to trigger long term plasticity of firing rates as suggested by Xu and Edgely?

---

## [Author Response]

Essential revisions:1) In several instances the authors conclude that climbing fiber activity provides a biphasic (excitatory followed by inhibition) control over Golgi cell spiking. However, this statement is mainly true in the presence of the glutamate transport blocker, TBOA. In the absence of TBOA, roughly half of the cells show a purely excitatory response (Figure 5). The authors need to acknowledge that the statement of biphasic response was only true in a subpopulation of cells, and discuss the caveats of these methodological details in terms of their interpretation of results in the broader context of cerebellar function. How do the authors imagine that these mechanisms are engaged under physiological circumstances, i.e. without TBOA/Picro/CGP, etc?

We agree that about half of the cells show a purely excitatory response, which is why we separated the two response types in Figure 5 recorded without TBOA (subsection “CF stimulation can suppress GoC firing”). As mentioned in the discussion, we think that heterogeneity in receptor activation profiles may contribute to the ability of GoCs to exhibit variable activity patterns necessary to distinguish between stimuli. To clarify this point, we have now modified the abstract to highlight these two types of responses.

It is important for us to clarify that these two response patterns should not be interpreted as defining the only two possible GoC response patterns in vivo. Rather, we propose that AMPA-receptor mediated depolarization and mGluR2-mediated inhibition are two distinct mechanisms that mediate direct control of GoC activity by CFs that can be combined to generate a variety of GoC activity patterns. As described further in Point 3 (below), our experimental methodology of stimulating CFs near GoCs “selects” for GoCs wherein excitation dominates, since all our GoC recordings show an AMPA EPSC. We speculate that GoCs in vivocan also respond solely with mGluR-mediated inhibition, but an experimental caveat is that we would not have identified GoCs with only inhibition due to the small mGluR current that is not readily apparent in voltage clamp. We have clarified this critical interpretive point in the Discussion section.

Inclusion GABA-receptor blockers was experimentally necessary to strictly confirm CF identity based on characteristic properties of the EPSC, but we speculate that CF excitation of GoC activity in vivoresults in an additional source of CF-mediated suppression of GoC spiking through feed-forward GABAergic inhibition (Discussion section). Finally, we added a discussion of the possible scenarios that may engage distinct CF-GoC activity patterns during in vivo(Discussion section).

2) Could the pause be driven by the higher peak spiking probability that occurs in TBOA? In Figure 4, the example shows a peak spiking probability of ~0.3 which increases to ~0.8. This ambiguity could be ruled out by calculating PSHs with different current injections that mimic the increases in peak probability and comparing the post-peak pauses.

As suggested, we have now included an experiment that shows the duration of GoC pause in spiking in response to current injections is indeed related to the initial spike frequency (new Figure 6—figure supplement 2). This result replicates a previously published report that the post-spiking “reset” of GoC firing frequency increases with the number of evoked APs (Solinas et al., 2007). However, we show that blocking mGluR2 receptors has no effect on this post-spike resetting whereas blocking mGluR2 receptors selectively abolished the pause following CF stimulation *without altering the peak firing frequency* (Figure 6). In addition, we showed that CF-mediated cumulative ratio is correlated with the mGluR2 current (Figure 9). Thus, while GoCs do exhibit post-injection spike “reset” related to spike frequency (potentially similar to what we showed for PF and MF spike resetting in Figure 7), the mGluR2 sensitivity suggests that CF-mediated pausing results from TBOA-enhancement of mGluR activation. This explanation has been included in the subsection “Stimulation of GoC afferents perturbs intrinsic firing by distinct mechanisms”.

3) In vivo recordings of Golgi cells following climbing fiber stimulation do not show excitation and always have a pause in firing (Xu and Edgley, 2008), whereas these data show a consistent excitation and a variable pause in firing. The authors should address this discrepancy between the in vivo and in vitro data.

We agree it is very important to address this point more clearly, since it relates to experimental limitations of both ex vivoand in vivoexperiments (also raised in Point 1). We stimulated CFs near GoCs and identified CF-EPSCs by well-known characteristics of CF spillover transmission (all-or-none EPSCs with PPD and TBOA sensitivity). Thus we “selected” only GoCs that show AMPA receptor EPSCs – these recordings could be viewed as similar to a “paired” recording between a single CF and a single GoC. in vivoexperiments use a different protocol consisting of stimulation of the inferior olive, a site that is distant from recorded GoCs. We speculate that the discrepancy between the in vivo and in vitro results reflects the difference in the probability of identifying a “paired” recording between a particular CF and GoC in the two preparations. We expect that CFs can generate a range of responses in GoCs including excitation only, biphasic excitation-inhibition and inhibition only, depending on the complement of glutamate receptors responding to spillover. Since mGluRs have a higher affinity for glutamate than AMPARs, it is possible that most GoCs in vivo exhibit solely an inhibitory response, but due to our selection criteria for confirming the CF source of glutamate, all of our GoC recordings have AMPA receptor components that will undoubtedly cause excitation. We have included this in the discussion as a possible explanation for the discrepancy between our data and prior in vivorecordings (subsection “Stimulation of GoC afferents perturbs intrinsic firing by distinct mechanisms”).

4) The rationale for this study is a bit overstated. In particular, it is not clear how the results of the present manuscript provide more than additional examples of how high release synapses such as CFs can access distant receptor pools, including mGluRs. Further, how the findings increase our understanding of how extrasynaptic signaling contributes to local circuit processes could be clarified. For example, does spillover transmission to mGluRs combine with MF inputs to trigger long term plasticity of firing rates as suggested by Xu and Edgely?

We agree that the introduction overstated the rationale for our study, since here we show for the first time that CF-stimulation can excite GoCs but we do not show how this excitation contributes to local circuit processes. We have toned down the introduction to more accurately describe our results (Introduction).

In response to this suggestion, we have added an additional section in the discussion that speculates the implications of this signaling in the cerebellar circuit, including its potential role in plasticity (subsection “Implications for granule cell processing and heterogeneity”).